# The Emerging Roles of γ-Glutamyl Peptides Produced by γ-Glutamyltransferase and the Glutathione Synthesis System

**DOI:** 10.3390/cells12242831

**Published:** 2023-12-13

**Authors:** Yoshitaka Ikeda, Junichi Fujii

**Affiliations:** 1Division of Molecular Cell Biology, Department of Biomolecular Sciences, Faculty of Medicine, Saga University, 5-1-1 Nabeshima, Saga 849-8501, Japan; 2Department of Biochemistry and Molecular Biology, Graduate School of Medical Science, Yamagata University, 2-2-2 Iidanishi, Yamagata City 990-9585, Japan

**Keywords:** cysteine, glutamic acid, L-5-oxoproline, γ-glutamylcyclotransferase, calcium-sensing receptor

## Abstract

L-γ-Glutamyl-L-cysteinyl-glycine is commonly referred to as glutathione (GSH); this ubiquitous thiol plays essential roles in animal life. Conjugation and electron donation to enzymes such as glutathione peroxidase (GPX) are prominent functions of GSH. Cellular glutathione balance is robustly maintained via regulated synthesis, which is catalyzed via the coordination of γ-glutamyl-cysteine synthetase (γ-GCS) and glutathione synthetase, as well as by reductive recycling by glutathione reductase. A prevailing short supply of L-cysteine (Cys) tends to limit glutathione synthesis, which leads to the production of various other γ-glutamyl peptides due to the unique enzymatic properties of γ-GCS. Extracellular degradation of glutathione by γ-glutamyltransferase (GGT) is a dominant source of Cys for some cells. GGT catalyzes the hydrolytic removal of the γ-glutamyl group of glutathione or transfers it to amino acids or to dipeptides outside cells. Such processes depend on an abundance of acceptor substrates. However, the physiological roles of extracellularly preserved γ-glutamyl peptides have long been unclear. The identification of γ-glutamyl peptides, such as glutathione, as allosteric modulators of calcium-sensing receptors (CaSRs) could provide insights into the significance of the preservation of γ-glutamyl peptides. It is conceivable that GGT could generate a new class of intercellular messaging molecules in response to extracellular microenvironments.

## 1. Introduction

Proteins and peptides are produced via the translation of genetic codes found in mRNA by ribosomes; some small peptides, however, are produced solely through sequential enzymatic reactions that depend on substrate specificity, but are independent of the genetic information encoded in the nucleotide sequences of DNA. These small peptides may contain non-protein amino acids or promote non-canonical peptide bonds between amino acids. Glutathione is the most abundant peptide of this category and exhibits pleiotropic functions [1,2]. Glutathione, in its reduced form (GSH), is produced via the coordinated action of γ-glutamyl-cysteine synthetase (γ-GCS) and glutathione synthetase (GS) without referencing genetic information. γ-GCS first ligates the γ-carboxyl group of L-glutamate (Glu) to the amino group of L-cysteine (Cys), which results in the production of L-γ-glutamyl-L-cysteine (γ-Glu-Cys) [3]. GS then adds a glycine (Gly) unit to γ-Glu-Cys, which leads to the formation of the tripeptide γ-Glu-Cys-Gly (i.e., GSH). Due to its broad substrate specificity, however, γ-GCS could be the producer of many other γ-glutamyl peptides, which are largely dependent on the availability of free amino acids.

The liver is the central organ that predominantly synthesizes GSH and secretes it into the bloodstream (Figure 1). It is generally accepted that glutathione, whether in its intact form, as conjugates, or as an oxidized dimer designated as glutathione disulfide (GSSG), is secreted from cells via the multidrug resistance-associated protein (MRP), which is a protein in the ABCC subclass of the ABC transporter family [4]. In circulation, the γ-glutamyl group appears to act as a protector against the peptidases that are abundant in blood plasma [5]. Some cells, however, express γ-glutamyltransferase (GGT) on the plasma membrane, which is involved in the initiation of glutathione degradation outside cells and helps recruit constituent amino acids [6]. GGT either hydrolytically removes the γ-glutamyl moiety of glutathione or produces a variety of γ-glutamyl peptides by transferring the γ-glutamyl moiety to an amino acid or dipeptide; these processes depend on the abundance of acceptor molecules [7].

Although both γ-GCS and GGT are responsible for producing γ-glutamyl peptides, since their discovery, the roles of the resultant γ-glutamyl peptides in vivo have remained unclear. The role of γ-glutamyl peptides, such as glutathione, as systemic allosteric modulators of calcium-sensing receptors (CaSRs) expressed on the plasma membrane, has attracted much attention, particularly in the fields of food chemistry [8,9] and neuroscience [10]. Since the production of γ-glutamyl peptides is elevated under certain pathological conditions, it is conceivable that they transmit signals from damaged cells to surrounding cells via the modulation of CaSR action. Here, we briefly overview glutathione function and its metabolism, with a particular focus on GGT. We then discuss the potential roles of the γ-glutamyl peptides that are extracellularly produced by GGT, which are compounds that have received scant attention so far.

## 2. Overview of Glutathione Function

The pleiotropic functions of glutathione are mostly associated with the sulfhydryl (SH) group in the Cys residue [1,2]. Glutathione is a major detoxification system for conjugating xenobiotic compounds [11,12], and also is instrumental in maintaining the redox state of cells by acting as an endogenous antioxidant [13,14]. Details of the phenotypic abnormalities and other information regarding human and mutant mice that carry defected glutathione-metabolizing enzymes are available in the corresponding literature [15,16,17].

### 2.1. Glutathione Conjugation in Xenobiotic Metabolism and in the Production of Intrinsic Bioactive Compounds

The conjugation of xenobiotics with GSH is catalyzed by glutathione S-transferase (GST), and, along with glucuronidation and sulfation, this process constitutes a major detoxification system in the liver and kidney [12,18]. The conjugated compounds are then excreted mainly to urine through the vascular system. Glutathione conjugates experience removal of the γ-glutamyl moiety via either GGT on the renal brush-border membrane or through other GGT-expressing cells such as the vasculature, which is followed by the removal of Gly units via extracellular dipeptidases (Figure 2). Such hydrolytic removal of amino acid moieties could confer new functions to the conjugates, as exemplified by the nephrotoxic action of cysteinyl acetaminophen (APAP) [19]. This mechanism also produces 5-S-cysteinyl dopamine in the brain, where it exerts neurotoxic activity [20].

Glutathione conjugation reactions could also be utilized to build the bioactive compounds observed in the process of synthesizing cysteinyl leukotrienes (CysLT) such as LTC4, LTD4, and LTE4. CysLTs are inflammatory lipid mediators involved in the pathophysiology of respiratory diseases [21] and are active components of the slow-reacting substances of anaphylaxis through the contraction of smooth muscle [22]. GST family members (MGST2 and 3, and GSTM4), as well as LTC4 synthetase, catalyze the conjugation of GSH to arachidonate epoxide, which is formed by 5-lipoxygenase [23]. Either GGT1 or GGT5 hydrolytically removes the γ-glutamyl group of LTC4 in the extracellular milieu, which results in LTD4. Removal of the Gly unit from LTD4 via dipeptidase finally results in the production of LTE4. This particular CysLT binds to the G-protein-coupled receptor subtypes CysLTR-1 and CysLTR-2 that are present in neurons, astrocytes, microglia, and vascular endothelial cells in the brain [24]. CysLT also binds to GPR17, GPR99, and PPARγ and could be involved in inflammatory responses.

### 2.2. The Role of GSH in Maintaining Redox Homeostasis

The essential roles that glutathione plays in maintaining cellular redox homeostasis are postulated to occur in most types of cells. The reactivity of GSH towards reactive oxygen species (ROS) is marginal and, in fact, the antioxidant action is largely accomplished by donating electrons to glutathione peroxidase (GPX) [25]. GPX uses two electrons from two GSH molecules to achieve the reductive detoxification of peroxides, which are oxidized into GSSG. While mammals carry eight genes encoding canonical GPX, proteins such as GST [26] and peroxiredoxin [27] also show GSH-dependent peroxidase activity to some extent.

Until the discovery of ferroptosis, it was unclear whether a GSH deficiency could be closely associated with a specific type of cell death, although the association of decreased GSH levels with oxidative stress-related cell death is implied in many diseases [28]. Ferroptosis is an iron-dependent form of regulated necrosis that was first discovered in cultured cells, wherein the function of xCT, a transporter of oxidized Cys referred to as cystine, is inhibited by erastin [29]. Glutathione levels are decreased via xCT inhibition due to the deprivation of cellular Cys, and enzymes that require GSH are disabled. Among the GSH-requiring enzymes, GPX4 reduces phospholipid hydroperoxide to corresponding alcohol and is one of the most potent enzymatic suppressors of ferroptosis [30,31]. Iron plays the primary role in the lipid peroxidation reaction, so that iron chelation also effectively suppresses ferroptosis. Under conditions with disabled GPX4, lipid peroxidation products accumulate, which leads to membrane rupture [32]. Because ferroptosis is considered to be involved in a variety of diseases, such as neurodegenerative disease, ischemic disease, and cancer [33], the primary role of the Cys-GSH-GPX4 axis in combating ferroptosis has begun to attract much attention. Although this issue is extremely important from the aspect of glutathione function, an in-depth discussion of ferroptosis is beyond the scope of this article. Therefore, interested readers should refer to the review articles dedicated to this issue [33,34,35,36,37,38].

Glutaredoxin, also called thioltransferase, is encoded by the gene GLRX and protects the SH groups in the Cys residue in proteins from oxidative modification via a reduction in the reducing equivalent of GSH [14,39]. Cysteine sulfhydryl (Cys-SH) forms four oxidation states: cysteine sulfenic acid (Cys-SOH), cysteine sulfinic acid (Cys-SO_2_H), cysteine sulfonic acid (Cys-SO_3_H), and a disulfide. While Cys-SOH and the disulfide can be reversibly reduced to Cys-SH via the use of either physiological reductants or enzymatic reactions, Cys-SO_2_H and Cys-SO_3_H generally cannot be reduced back to Cys-SH via biological systems [40]. Under oxidative conditions, glutathione forms a conjugation with Cys-SH and more preferentially with Cys-SOH in proteins, which is called S-glutathionylation [41]. While Cys-SOH is prone to oxidation to either Cys-SO_2_H or Cys-SO_3_H, such oxidation is avoided by competitive mixed disulfide formation with glutathione. When cells are recovered from oxidative stress, S-glutathionylated Cys residue can be reduced back to Cys-SH residue by means of another GSH, which is accelerated via the catalytic action of GLRX. Thus, S-glutathionylation could prevent the irreversible oxidation of proteins under oxidative conditions. GLRX is encoded by two genes in mammals: GLRX1 and GLRX2. While GLRX1 resides mainly in the cytoplasm, GLRX2 is localized either in the mitochondria or nuclei [42,43].

## 3. Maintenance of Intracellular Glutathione

Cellular Cys content is kept low as it is cytotoxic in high concentrations and, hence, Cys availability occasionally determines the amount of glutathione synthesis. Most cells, even red blood cells lacking a protein-synthesizing system, produce glutathione from Glu, Cys, and Gly via the sequential reactions of γ-GCS and GS. Cys is either imported from outside the cell in the form of free Cys/cystine or it is synthesized through the trans-sulfuration pathway in conjunction with L-methionine (Met) metabolism in competent cells [44].

### 3.1. Extracellular and Intracellular Cys Sources

Because Cys availability may limit glutathione synthesis, in order to elucidate glutathione metabolism, it is important to understand the unique mechanisms for maintaining cellular Cys levels. Animals are heterotrophs and cannot synthesize approximately half of the amino acids required as building blocks for proteins. In high concentrations, however, some amino acids can be toxic to certain cells, as observed with Glu toxicity in neurons [45] and Cys toxicity in many types of cells, including neurons and hepatocytes. Rather than Glu, it is glutamine (Gln) that is the dominant form in blood. Similarly, to avoid such risk, it is glutathione that may be circulating rather than Cys. However, to recruit Cys from glutathione, the γ-glutamyl moiety must first be removed outside the cell, and this removal is performed by GGT. Extracellular dipeptidase then degrades the Cys-Gly dipeptide and releases Cys as well as Gly. Neutral amino acid transporters (NAATs), typically the alanine serine cysteine transporter (ASCT) and the L-type amino acid transporter 2 (LAT2), mediate the cellular uptake of Cys in ordinary cells [46]. In neurons, neutral amino acid transporter excitatory amino acid carrier type 1 (EAAC1, also called EAAT3) plays essential roles in Cys uptake [47].

Under oxidative conditions, Cys is oxidized to cystine, which is the predominant form in the extracellular fluid of the body and in cultivation media. There are two distinctive transport systems for cystine: system b^0,+^ and system x_c_^−^. While b^0,+^ AT, also referred to as SLC7A9, is the main component for system b^0,+^ and is expressed exclusively in the kidneys, xCT, which is referred to as SLC7A11, plays the main role in system x_c_^−^ [48,49]. Cystine taken up by cells is then reduced to Cys by TRX1 and TRX-related protein14 kDa (TRP14) [50]. Under normal conditions, xCT shows no structural similarity to b^0,+^ AT and is constitutively expressed only in certain organs such as the brain and the immune system. However, the expression of xCT can be induced in many types of cells in response to stress conditions such as oxidative stress and hyperoxia, which are likely mediated by transcriptional regulatory factors Nrf2 and ATF4 [51,52]. Cystathionine is also taken up by xCT [53] and can be utilized to synthesize Cys via the trans-sulfuration pathway in competent cells. Because cystine is dominant in cell culture media, xCT expression is commonly induced in most types of cells, whether lined or primary, to meet the Cys requirement and support their survival [48].

Most non-essential amino acids are synthesized by transferring an amino group from Glu to the carbon backbone, α-keto acid, of the corresponding amino acid. However, Cys is the result of transferring a sulfur atom that originates from Met to the main component of L-serine (Ser), which is why the process for Cys formation is referred to as a trans-sulfuration reaction [44]. In Met metabolism, the reaction of Met and ATP first produces S-adenosylmethionine (SAM), which acts as a methyl group donor for various compounds and is converted to S-adenosyl-homocysteine after the transfer of the methyl group. Adenosine is then released from S-adenosyl-homocysteine to form L-homocysteine. Cystathionine β-synthase combines L-homocysteine with Ser to form cystathionine. Finally, cystathionine γ-lyase cleaves cystathionine to Cys and 2-oxobutyric acid. Since the sulfur in Cys-SH originates from Met, Cys production depends on the content of Met and on the metabolic activities of the trans-sulfuration pathway. Therefore, it is not only the content of Cys, but also Met availability, that limits glutathione synthesis in some cases. The trans-sulfuration pathway coupled with Met metabolism is highly active in hepatocytes and in other cells, which must meet the demand for Cys.

### 3.2. De Novo Glutathione Synthesis

γ-GCS is responsible for the production of γ-Glu-Cys, and then, GS completes glutathione formation via the ligation of a Gly unit. The enzymatic and protein chemical properties of γ-GCS have been precisely reviewed by Griffith [3], so the subject matter is only briefly described here. γ-GCS is a heterodimeric enzyme composed of a catalytic subunit that is encoded by the GCLC gene and a modifier subunit that is encoded by the GCLM gene [15]. Physiological levels of GSH suppress γ-GCS activity in an allosteric manner. The GCLC protein is responsible for catalysis and is regulated by a specific inhibitor, buthionine sulfoximine, which decreases glutathione to negligible levels after one day of treatment in most types of cells during cultivation [54]. This indicates that γ-GCS activity is the main component for maintaining glutathione levels inside cells.

The generation of γ-Glu-Cys by the γ-GCS-catalyzed ligation reaction proceeds essentially in two partial reaction steps [55]. In the first step, Glu is phosphorylated by ATP to form a carboxylic phosphoric anhydride, γ-glutamyl phosphate. Then, this activated γ-carboxyl group is attacked by the amino group of Cys to form γ-Glu-Cys. Cys is the preferable amino acid substrate that is γ-glutamylated in the γ-GCS reaction (the Michaelis constant (Km) = 0.1–0.3 mM), but Cys-mimetic 2-aminobutyric acid (2AB) is also a good substrate [3]. Moreover, other amino acids also take the place of Cys, albeit to a lesser extent, as described in the following section. This broad specificity to the substrate allows for the production of a variety of γ-glutamyl peptides inside cells. Mice with a genetic ablation of GCLC exhibit embryonic lethality [56]. While at least six children with GCLC deficiency are reported to have shown anemia [57], significant levels of glutathione remained in their red blood cells, which suggests the preservation of some γ-GCS activity.

GS is a homodimeric enzyme that adds a Gly unit to γ-Glu-Cys and also to γ-Glu-2-aminobutyrate (2AB), which results in GSH and γ-Glu-2AB-Gly, which is referred to as ophthalmic acid (OPT), respectively. A genetic deficiency of GSS encoding GS is quite rare and shows a relatively mild phenotype [58,59]. GSS-deficient patients experience 5-oxoprolinuria, hemolytic anemia, and neurological dysfunction. Whereas glutathione levels are decreased in the fibroblasts of GSS-deficient patients, Cys and γ-Glu-Cys levels increase [60]. Hemolytic anemia and neurological dysfunction can be explained as a decrease in glutathione levels. The excessive production of L-5-oxoproline (5-OP) causes 5-oxoprolinuria, and Cys sometimes accumulates inside the cells of patients with a GSS deficiency. In the case of the γ-GCS reaction, the generation of 5-OP and the accumulation of Cys are contradictory phenomena because 5-OP may be produced when γ-glutamyl phosphate does not meet Cys [55]. Another hypothetical mechanism involves a certain isozyme of γ-glutamylcyclotransferase, which produces 5-OP from glutathione and could also actively cleave γ-Glu-Cys to 5-OP and Cys. Currently, however, there is no evidence to support either mechanism.

### 3.3. Reductive Recycling of GSSG

The production of GSSG is a result of one of the following: the non-enzymatic oxidation of GSH, electron donation to the enzymatic reaction of GPX, or thiol transfer from a glutathione-mixed disulfide via GLRX. Glutathione reductase (GSR) uses NADPH as the primary electron donor to catalyze the reductive recycling of GSSG to two GSH [61], and a portion of GSSG is transported out of cells via MRP. GSR is constitutively present in most cells, but can be induced in an Nrf2-dependent manner under oxidative stress, and it plays a crucial role in preserving the cellular redox state via the regeneration of GSH [62].

NADPH is the primary electron source for the GSR reaction as well as for many other reductase reactions, and maintaining its levels is extremely important for redox homeostasis and its support of many anabolic reactions, such as the reductive conversion of ribonucleotides to deoxyribonucleotides and lipogenesis [63]. NADPH is partially produced by accepting electrons from NADH through nicotinamide nucleotide transhydrogenase in the mitochondria of ordinary cells [64], but the capacity is limited. Glucose-6-phosphate dehydrogenase (G6PD) in the pentose phosphate pathway is the dominant enzyme that produces the required amount of NADPH during the conversion of glucose-6-phosphate to D-glucono-1,5-lactone-6-phosphate [63]. Increased G6PD activity appears to meet the demand of NADPH for cell proliferation and for combating oxidative stress. GSH is abundantly synthesized in red blood cells, and the reductive recycling of GSSG by GSR is particularly important for prolonging their life span. A defect in G6PD is the most prevalent form of enzyme deficiency in the world and is closely associated with anemia [65]. A proposed mechanism for developing anemia from a G6PD deficiency involves the premature removal of red blood cells from the circulation due to a decrease in deformability. When GPX is disabled due to a defect in glutathione recycling, the redox state of red blood cells shifts toward an oxidative stage, which impairs deformability. Patients with a G6PD deficiency are generally asymptomatic in ordinary life, but stimuli that cause oxidative stress, such as microbial infections and the use of certain medications, could trigger the development of anemia.

## 4. GGT, a Key Enzyme in Extracellular Glutathione Metabolism

There is a group of cell-penetrating (permeable) peptides that traverse the phospholipid bilayer barrier without the involvement of a transporter protein [66]. However, due to its hydrophilic nature, GSH does not follow this process. While S. cerevisiae import GSH into the cells via Hgt1p in a proton-coupled manner [67,68], no orthologous genes are known to exist in mammals. Although mitochondrial GSH import is mediated by a probable transporter protein, SLC25A39 [69], neither GSH nor related γ-glutamyl peptides appear to enter mammalian cells without modification, e.g., via esterification [70].

Glutathione is actively degraded on the surface of some types of cells under physiological and pathophysiological situations (Figure 3). Extracellular GSH and GSSG experience removal of the γ-glutamyl moiety by GGT, which is a plasma membrane-anchored glycoprotein, and are further hydrolyzed into amino acids by dipeptidase [7]. A similar degradation process is also involved in the metabolism of glutathione S-conjugate with an aromatic compound, which is finally metabolized to mercapturic acid via the N-acetylation of the cysteine S-conjugate, and is then excreted from the body through the urine. The human GGT gene family, which includes GGT-related genes along with pseudogenes, has been classified by Heisterkamp et al. [71] in collaboration with the HUGO Gene Nomenclature Committee (HGNC). We adopted the proposed nomenclature in this article.

### 4.1. γ-Glutamyltransferase

The reaction of glutathione with amino acids to yield peptides containing glutamic acid was first described by Hanes et al. in 1950 [72], and this led to the observation that an enzyme from the kidney catalyzes removal of the γ-glutamyl group from glutathione and the formation of a new γ-glutamyl bond with various acceptor amino acids, and thereby produces several types of γ-glutamyl amino acids. Thus, the enzyme was referred to as γ-glutamyl transpeptidase (γ-GTP) [73] and thereafter as γ-glutamyltransferase (GGT). The hydrolytic reaction was also found to be catalyzed by the same enzyme, which transfers a γ-glutamyl group to a water molecule. Mammalian GGT also exhibits glutaminase activity, by which glutamine is hydrolyzed to glutamic acid and ammonia, as indicated by the identification of GGT as maleate-stimulated glutaminase [74,75].

Mammalian GGT is a heterodimeric type-II transmembrane glycoprotein that is translated as a single precursor and post-translationally processed into a dimeric form by autocatalytic limited proteolysis [76]. The enzyme protein is anchored to a cell surface, and can be released into various body fluids via cleavage of the membrane-anchor domain and/or other non-hydrolytic processes [77,78,79,80].

GGT has been a traditional biomarker of hepatobiliary diseases, and recently has also been considered a potential biomarker for various other diseases [81]. Measurements of the blood levels of enzyme activity are of particular clinical importance in the diagnoses of liver diseases such as alcoholic damage and carcinogenesis. Many studies have accumulated evidence that this enzyme is induced by the administration of rugs and xenobiotics, including ethanol, and during carcinogenesis in various tissues. Elevation of the enzyme activity in the blood seems to be caused either by the increased expression of enzyme protein in lesions or by enhanced liberation from cells—and possibly by both of these factors. In clinical chemistry testing, various “GGT iso(en)zymes” have been observed in electrophoretic separation and have been investigated to improve diagnostic accuracy [79]. The appearance of different forms of GGT in clinical chemistry testing appears to be attributed to variation in the N-terminal structure, which is due to non-specific limited proteolysis as well as to an association with exosomes [80] and/or simply to the structures of glycans [82].

### 4.2. Substrate Specificities and Enzyme Assay of GGT

GGT uses a broad range of γ-glutamyl compounds as donor substrates as well as glutathione [83]. Various γ-glutamyl derivatives of amino acids and other natural and synthetic compounds serve as γ-glutamyl donor substrates. In most popular and general assays for GGT, γ-glutamyl derivatives of 4-nitroaniline are often used as the donor to allow for spectrophotometric measurements of enzyme activity [84]. With respect to acceptors, the substrate specificity of the enzyme is somewhat broad. The enzyme efficiently transfers the γ-glutamyl moiety to some amino acids and dipeptides, whereas glycylglycine is a relatively good acceptor substrate, and it most often serves as the acceptor. Because L-γ-glutamyl-p-nitroanilide is disadvantageous for practical use in clinical chemistry testing due to its slight solubility, L-γ-glutamyl-3-carboxy-4-nitroanilide, an alternatively modified substrate, is instead generally favored for easier handling in clinical examinations [85].

### 4.3. Enzyme Reactions of GGT: Transpeptidation, Hydrolysis, and Autotranspeptidation of γ-Glutamyl Compounds

In addition to the transpeptidation reaction, GGT also catalyzes the hydrolysis of the γ-glutamyl bonding of γ-glutamyl substrates. Furthermore, when glutamine is used as the substrate, the enzyme hydrolyzes the side chain amide group, yielding glutamic acid and ammonia. All these reactions certainly appear to involve an acyl enzyme intermediate, which is the γ-glutamyl enzyme. The intermediate is subsequently attacked by amino groups of the acceptors or by water, and, thus, produces either new γ-glutamyl compounds or glutamic acid, respectively. The transfer reaction of the γ-glutamyl moiety could be considered GGT-mediated re-distribution and propagation of the γ-glutamyl group among various biological molecules in organisms.

When γ-glutamyl-p-nitroanilide is used as the substrate for hydrolysis, the formation step of the acyl enzyme is faster than the step for the water-conducted breakdown of the intermediate, as shown by the faster reaction rate for the transpeptidation reaction compared with that of hydrolysis [86]. The nucleophilic attack of an amino group toward the enzyme intermediate is more efficient than that of water. However, since the simultaneous hydrolysis is not necessarily marginal compared with that of transpeptidation, researchers have long questioned whether the biochemical significance of the enzyme is attributed to transpeptidation or to the hydrolysis of γ-glutamyl compounds that include glutathione [87].

In addition to the usual transpeptidation and hydrolysis of the γ-glutamyl group, GGT also catalyzes autotranspeptidation, which is an unusual type of transpeptidation [88]. When the enzyme is allowed to react with a γ-glutamyl donor in the absence of an acceptor substrate, the γ-glutamyl group is transferred to another molecule of the γ-glutamyl donor, which results in the formation of a γ-glutamyl-γ-glutamyl compound and/or a γ-glutamyl-γ-glutamyl-γ-glutamyl compound [89]. Autotranspeptidation occurs only when using the L-stereoisomer of the γ-glutamyl group rather than the D-isomer, which is consistent with evidence that the D-isomers of amino acids do not serve as an acceptor for the reaction [90]. When the D-γ-glutamyl donor is used as a substrate, the enzyme exhibits only hydrolysis and not autotranspeptidation [86]. When the hydrolytic activity is assessed using a L-γ-glutamyl donor without any acceptor substrate, it should be noted that the appropriate concentration of the donor must be chosen. The concentration of the donor should be sufficiently low for the transpeptidation activity to be negligible. In the absence of an acceptor, a double reciprocal plot for the donor substrate shows a “substrate activation” profile rather than a straight curve, as indicated by the downward curvature with a decreasing 1/[substrate] [86,91]. The autotranspeptidation becomes noticeable as the substrate concentration increases, because the apparent Km value for the single substrate as an acceptor is much higher than its value as a donor.

### 4.4. Protein Structure, Autoprocessing, and the Active-Site Chemistry of GGT

The crystal structures of human GGT have been solved both in various forms complexed with several inhibitors and other ligands, as well as in a ligand-free form [92,93,94,95]. These analyses indicate that GGT contains an α-β-β-α core structure, as found in the N-terminal nucleophile (Ntn) hydrolase superfamily. Three sides of the small subunit of the human GGT are surrounded by the large subunit, the structure of which appears as a hand wrapped around a ball. A longitudinal crevice exists on the exposed surface of the small subunit and forms a substrate channel (Figure 4).

Ntn hydrolases are translated as inactive precursors and are then activated by autoproteolytic processing into a mature form. In human GGT, autocatalytic activation involves an N-O acyl shift at threonine (Thr)-381, which produces an ester bond between the side chain OH group of the Thr and an α-carbonyl group of the residue 380, which is the last residue of the N-terminal portion that consequently becomes the large subunit. Hydrolysis of the ester bond results in formation of the heterodimeric active form of GGT. The OH group of the Thr-381, which is the most N-terminal residue of the small subunit after processing, serves as a catalytic nucleophile in the active center to form an acyl enzyme intermediate, which is the so-called γ-glutamyl enzyme, during the catalytic process of the enzyme reaction.

The nucleophilic attack of the hydroxyl group of Thr-381 against the 5-carbonyl carbon of the γ-glutamyl moiety is facilitated by the α-amino group of the same residue, which acts as a general acid–base catalyst generally found in the Ntn-hydrolase superfamily (Figure 5A). The crystal structures of inhibitor-complexed forms also indicate that Gly-473 to Thr-475 forms an oxyanion hole, which interacts with the oxyanion of the tetrahedral intermediate in the transition state. Peptide bond nitrogen atoms of Gly-473 and Gly-474 appear to form hydrogen bonds with the negatively charged oxygen atom, thereby stabilizing the transition state (Figure 5B).

A γ-glutamyl group of the substrates is recognized by several residues of the active site in human GGT (Figure 6). The α-amino group of the γ-glutamyl moiety interacts with the carboxyl groups of Glu-420 and aspartate (Asp)-423 and with the amide carbonyl group of asparagine (Asn)-401 via electrostatic and hydrogen bonding. On the other hand, the α-carboxyl group interacts with the guanidino group of arginine (Arg)-107 and with the hydroxy group of Ser-451. The carboxy group of the substrate also forms hydrogen bonds with the peptide nitrogen of Ser-452 and possibly its side chain hydroxy group. These interactions allow the enzyme to specifically recognize the γ-glutamyl group. Because GGT tolerates the D-stereoisomer of the γ-glutamyl group as the donor, ionic bonds may be more significant than hydrogen bonds due to their nondirectional characteristic.

### 4.5. Gene Family of γ-Glutamyltransferase

Although GGT had long been considered a unique enzyme with γ-glutamyl bond-cleaving activity, particularly with respect to glutathione metabolism, another closely related but clearly different enzyme, referred to as the “GGT-related enzyme (GGT-rel)”, is known to catalyze the hydrolysis of glutathione [96]. Further studies have identified several genes with nucleotide or amino acid sequences that are similar to the most “traditional” γ-glutamyl transferase, the gene symbol of which has been systematically designated as GGT1. Thus, these related genes constitute a gene family, which includes at least 13 members that have been concisely summarized by Heisterkamp et al. [71]. Of the 13 genes, at least 6 appear to be active in terms of expression. Nevertheless, the protein products of only two genes, GGT1 and GGT5, the latter of which is the approved gene symbol for GGT-rel, were found to function as enzymes. GGT5 is capable of hydrolyzing glutathione but not γ-glutamyl-p-nitroanilide, which is an ordinary substrate for enzyme-activity assay. It seems most likely that one of the most important roles of GGT5 is the conversion of leukotriene C4 into D4 by hydrolyzing the γ-glutamyl bond, and therefore, the enzyme is also known as γ-glutamyl leukotrienase [97]. In fact, GGT5 is primarily expressed in the spleen, and GGT5-deficient mice indicate a defect in the LTD4 formation and the attenuation of acute inflammatory responses [98,99]. By contrast, GGT1-null mice show a substantial conversion of LTC4 to LTD4, which excludes GGT1 from the responsible genes [97,100]. The enzymatic properties of GGT5, as well as those of GGT 7, have not been well characterized in terms of the formation of γ-glutamyl peptides.

Thirteen genes are systematically designated as GGT1 through GGT8 for the genes that comprise full-length whole proteins with both large and small subunits, and as GGTLC1 through GGTLC5 light-chain-only genes, wherein only the small subunit (light chain) is encoded [71]. Of the full-length protein genes, GGT3, GGT4, and GGT8 are not functional and are considered to be pseudogenes, thus being more exactly noted as GGT3P, GGT4P, and GGT8P, respectively. As described above, although GGT1 and GGT5 products have been well characterized in terms of enzymology, biochemistry, and structural biology, other full-length genes have not yet been investigated in sufficient detail. With respect to the light-chain-only genes, although GGTLC1 through GGTLC3 are functional genes, the roles of their protein products are unknown. Enzymatic activities such as the abilities of hydrolysis and the transpeptidation of γ-glutamyl compounds involve amino acid residues located in the large subunit, and therefore, it is very unlikely that these proteins exhibit the same or similar activities compared to those of γ-glutamyltransferase. On the other hand, GGTLC4 and GGTLC5 are pseudogenes, and should be noted as GGTLC4P and GGTLC5P, respectively.

### 4.6. Structure, Expression, and Deficiency of the GGT1 Gene

In the GGT1 gene, the coding region spans more than 16 kb and comprises 12 exons and 11 introns. Exons 1 through 7 encode the large subunit, while exon 8 encodes the carboxy portion of the large subunit and the amino terminal portion of the small subunit. Exons 9 through 12 encode the residual large portion of the small subunit [101]. Such gene organization is similar to that of mouse GGT genes [102]. Multiple promoters allow for tissue-specific and/or inducible expression and even for the production of various transcripts.

GGT deficiency is a rare disease, and probably fewer than 10 cases have been reported [103]. The disease is characterized by glutathionuria, which is due to the increased excretion of glutathione in the urine, low GGT activity in the serum, and higher levels of plasma glutathione. Some patients show mental retardation and neurological symptoms. In recent cases [103], whole-genome sequencing showed an approximately 17 kb-homozygous deletion in the GGT1 gene, which includes the first coding exon and several non-coding exons. Heterozygous parents and siblings of these patients are healthy with no symptoms, which clearly indicates an autosomal recessive disorder. Although GGT expression is highest in the kidneys and intestines but lower in the brain, the central nervous system discrepantly appears to be the most affected by the deficiency.

## 5. Enzymatic Reactions Involved in Intracellular Glutathione Metabolism

The glutathione-degrading system is not very active inside cells, but the inhibition of glutathione synthesis by treatment with a γ-GCS inhibitor reduces the content to quite low levels. This is thought to be due mainly to the excretion of GSH, glutathione conjugates, and GSSG by MRP. The degradation of glutathione, however, may be accelerated under conditions such as starvation [104] and endoplasmic reticulum (ER) stress [105].

### 5.1. GSH-Specific γ-Glutamylcyclotransferase Activity Inside Cells

Glutathione degradation inside cells is initiated by isozymes of γ-glutamylcyclotransferase (Figure 3). Initial studies have shown that γ-glutamylcyclotransferase liberates the γ-glutamyl moiety from γ-glutamyl amino acids, excluding GSH, as 5-OP [1]. While such activity has been recognized for a long time, the protein that exhibits glutathione-specific γ-glutamylcyclotransferase activity was only recently identified. The first identified enzyme was cation transport regulator protein 1 (CHAC1), which was originally reported as a proapoptotic component [105]. The CHAC family proteins consist of CHAC1 and CHAC2, and their genes are located on human chromosomes 15 and 2, respectively. CHAC specifically acts on GSH with a Km of 0.1–2 mM, which is the appropriate value for the degradation of GSH inside cells [106,107]. However, CHAC 1 shows activity that is approximately 20 times higher than that of CHAC 2.

Studies on CHAC 1 and CHAC 2 have shown that the two isoenzymes both promote the intracellular degradation of GSH, but they appear to act in different contexts. ChaC1 is upregulated under stress conditions such as ER stress and amino acid starvation [105], whereas ChaC2 is constitutively expressed and may act in a housekeeping fashion [5]. An ATF4-CHOP cascade is involved in the induction of ChaC1 under ER stress conditions, which results in the accelerated degradation of intracellular glutathione and may be responsible for cell death. ChaC1 is also induced by xCT inhibition and therefore is considered a potential marker for ferroptosis [108]. Since the GPX4-catalyzed reduction of lipid peroxides is disabled by glutathione deprivation, elevated ChaC1 may play a role in ferroptosis by degrading glutathione [109]. While the participation of both ATF3 and ATF4 has been proposed in the induction of ChaC1 [110], the cooperation of both FOXO1 and AFT4 reportedly also plays a role in the induction [111]. The administration of paracetamol, another name for acetaminophen (APAP), induces ChaC1, and the eIF2α-ATF4 pathway that is activated under ER stress appears to be involved in this induction [112]. In any case, ATF4 appears to have a primary role in the induction of the ChaC1 gene. ChaC1, which is referred to as Botch in the report [113], is induced and promotes neurogenesis by antagonizing Notch signaling in mouse models. The enzymatic reaction that has been proposed for ChaC1/Botch by Chi et al. [113] is the formation of 5-OP by deglycinase. This enzymatic reaction appears to be unlikely, however, because ChaC1/Botch exhibits γ-glutamylcyclotransferase activity but not deglycinase activity [5]. DJ-1 is the protein that protects against the development of Parkinson’s disease (PD), and its mutation causes early-onset PD in an autosomal recessive manner [114]. Because ChaC1 is upregulated in DJ-1 knockout mice, DJ-1 is considered to maintain glutathione by suppressing ChaC1 expression, which thereby prevents PD development [115]. Mice with a knock-in of inactive mutant ChaC1 tend to sustain glutathione levels in their muscles during starvation, but otherwise show no significant changes [116]. Since mutant ChaC1-knock-in mice have been examined only in limited situations, it remains ambiguous whether a deficiency of CHAC 1 activity affects phenotypic properties under pathological conditions such as ER stress, oxidative stress, and ferroptosis.

Unlike CHAC 1, CHAC 2 is enriched in undifferentiated human embryonic stem cells [117]. The downregulation of ChaC2 tends to decrease the levels of glutathione and blocks the self-renewal of these cells. Simultaneous knock-downs of ChaC1 and ChaC2, however, restore the self-renewability of the cells, which indicates that glutathione homeostasis is maintained by the balance between ChaC2 and ChaC1. ChaC2 levels are associated with some malignant diseases. For instance, ChaC2 expression is frequently downregulated in gastric and colorectal cancers, which suggests it is a tumor-suppressor gene [118]. However, the elevated expression of ChaC2 in breast cancer [119] and in hepatocellular carcinoma [120] is associated with a poor prognosis for these malignant diseases. While these observations are discrepant at a glance, a similar situation is sometimes observed in other antioxidant systems associated with tumors [121]. A possible explanation for such observations may be that antioxidant systems are originally protective against tumorigenicity through a reduction in mutagenic ROS. Once tumor cells develop and acquire a high antioxidant ability, however, they become resistant to radiation and chemotherapy. Since investigations of CHAC isozymes have only recently begun, further studies are required to reveal how they express differential functions in terms of glutathione metabolism in vivo.

### 5.2. Conversion of 5-OP to Glu by 5-Oxoprolinase

The 5-OP that is the result of CHAC-catalyzed glutathione cleavage is converted to Glu by 5-oxoprolinase (OPLAH) in an ATP-dependent manner [122]. Since the first report on the genetic deficiency of OPLAH [123], more mutations of the gene have been reported in patients who commonly develop 5-oxoprolinuria [124]. The ablation of OPLAH in mouse models has led to an accumulation of 5-OP, which results in oxidative stress, fibrosis, and the continued elevation of pressure due to heart failure [125]. The elevation of 5-OP is sometimes seen in patients taking APAP, which implies an association of anion gap metabolic acidosis with 5-oxoprolinuria, but no solid evidence has made a connection to OPLAH [126]. Further studies of mice engineered for CHAC and OPLAH that are involved in 5-OP metabolism may provide answers to this issue.

### 5.3. Hydrolysis of Cys-Gly Dipeptide

Glutathione is a dominant reductant in most cells and is also considered to be a vehicle for transporting Cys in a safe form to cells in vivo. GGT-catalyzed removal of the γ-glutamyl moiety from glutathione releases a Cys-Gly dipeptide, which undergoes either direct uptake by peptide transporter PEPT2 in some cells or further degradation into free amino acids, Cys and Gly, via extracellular dipeptidases [127,128]. Several dipeptidases appear to perform this reaction in the extracellular space, and carnosine dipeptidase (CNDP) 2 has been identified as a protein that preferentially reacts on the Cys-Gly dipeptide in yeast [129]. While CNDP1 specifically degrades anti-inflammatory dipeptide carnosine (β-alanyl-L-histidine) in blood plasma, CNDP2 is a cytosolic isoform and is more specific to Cys-Gly dipeptide in animals [130]. We found that the CNDP2 level was elevated in primary macrophages isolated from xCT-knockout mice [131]. Although the genetic ablation of CNDP2 does not cause a vast change in Cys-Gly dipeptidase activity, APAP overdose induces renal damage more severely than that encountered in wild-type mice. Therefore, CNDP2 along with GGT may facilitate the recruitment of Cys from extracellular GSH when demand increases during emergencies.

The risk of nephropathy increases in cases of type 2 diabetes that carry common variants of CNDP1 and CNDP2 genes [132]. Tumor suppressor action by CNDP2 has also been observed in hepatocellular carcinoma [133] and in gastric cancer [134]. Although the mechanism was not clarified in these studies, the resultant increase in GSH could enhance antioxidant capacity, thereby suppressing nephropathy development and tumorigenic mutagenesis. In contrast, the upregulation of CNDP2 has been reported in several tumors such as hepatocellular carcinoma [133]. CNDP2 may also stimulate the growth of colon cancer [135] and ovarian cancer cells [136] by recruiting Cys for the synthesis of GSH. Because ubenimex, also called bestatin, is an inhibitor of CNDP2, the anti-tumorigenic action of this drug could be partially attributed to the inhibition of CNDP2 [137]. It is also noteworthy that CNDP2 has a novel function in that it catalyzes N-lactoyl amino acid formation [138]. Exercise increases N-lactosyl phenylalanine in a CNDP2-dependent fashion, which may help control food intake and regulate systemic energy balance [139]. Thus, it is possible that elevated N-lactosyl amino acids produced by CNDP2 also have additional functions in tumorigenesis.

## 6. Physiological Significance of γ-Glutamyl Peptide Production

A variety of γ-glutamyl peptides are produced by γ-GCS- and GGT-involved reactions, but with the exception of glutathione, their functions in vivo remain largely unknown. An intriguing question is whether γ-glutamyl peptides produced by these different enzymes could have differential roles. Recent studies have provided clues to understanding them.

### 6.1. γ-GCS/GS Are Intracellular Producers of γ-Glutamyl Peptides

γ-GCS and GS are generally understood to work together for the purpose of glutathione synthesis, but uncoupling could occur between these enzymes. Indeed, γ-Glu-Cys is present in a fairly large amount, even in healthy mice, notably in the kidneys [140]. Regarding γ-Glu-Cys production, the proteolytic removal of Gly from glutathione by peptidase is an unlikely mechanism because prior removal of the γ-glutamyl group is required before removing the Gly unit [5]. It is also unlikely that GGT produces γ-Glu-Cys in significant levels due to the transfer activity for the γ-glutamyl group, because extracellular Cys concentrations are limited to serving as the substrates of γ-glutamyl reactions among free amino acids under physiological conditions. Thus, it is conceivable that γ-Glu-Cys is dominantly produced by the reaction of γ-GCS, per se, in some cells. Judging from the fact that γ-GCS and GS proteins do not associate and that their gene expression is controlled differently [141], it is reasonable to consider that in certain situations, γ-GCS acts dominantly compared with GS and produces γ-glutamyl peptides other than glutathione.

The synthesis of these γ-glutamyl peptides by γ-GCS and GS appears more likely to occur when cells are under pathological conditions that feature a Cys deficiency (Figure 7). Upon extreme consumption of glutathione, as typically observed in cases of APAP overdose, cellular Cys is consumed due to excretion as the APAP conjugates to urine [19]. As a result of Cys insufficiency, γ-GCS may promote the ligation of Glu to other amino acids via γ-glutamyl bonding, which results in the production of a variety of γ-glutamyl peptides. Hepatic damage with oxidative stress or excessive glutathione conjugation limits Cys, which could lead to the aberrant production of γ-glutamyl peptides, as reported in hepatic injury [142]. The 2AB that is formed when 2-oxobutyric acid accepts an amino group is abundant in the liver and is a good substrate for the γ-GCS reaction instead of Cys. Under a Cys deficiency, such as an APAP overdose, OPT is predominantly generated through a sequential reaction of γ-GCS and GS [143]. The elevated production of OPT is typically observed in APAP-overdosed mice [144], in patients with hepatic injury [142], and in fasted mice [145]. While the presence of physiological levels of GSH inhibits γ-GCS activity via an allosteric mechanism, OPT has no inhibitory effect [3], which, together with a low level of Km of 2AB for γ-GCS, leads to the abundant production of OPT among γ-glutamyl peptides under a Cys deficiency. Thus, γ-glutamyl peptides, notably OPT, could be considered as markers for liver damage in which a Cys deficiency becomes more pronounced through the glutathione conjugation reaction and oxidative stress.

### 6.2. Benefits of the Production of γ-Glutamyl Peptide by the γ-GCS-Involved Reaction Inside Cells

While beneficial actions of γ-Glu-Cys have been demonstrated in many studies [146,147,148], these are largely attributable to the reactivity of the Cys moiety. Typically, γ-Glu-Cys can also support GPX activity by directly donating electrons [149]. Cys is released from γ-Glu-Cys, and then, recruited to synthesize glutathione. However, other γ-glutamyl peptides do not have redox activity, so their production may have different underlying benefits. A list of the possible mechanisms follows, which could help elucidate this issue.

One hypothetical explanation for producing these γ-glutamyl peptides could be associated with the unique enzymatic properties of γ-GCS that cause a futile cycle in cases of Cys deficiencies. γ-Glutamyl phosphate is the intermediary compound produced from Glu and ATP by the action of γ-GCS; this compound is autocyclized to form 5-OP when there is an insufficient amount of Cys as a substrate, as discussed above in the development of 5-oxoprolinuria. The resultant 5-OP is converted back to Glu by OPLAH, which is a reaction that also requires ATP. Since these collective reactions merely consume ATP without producing substantial products, they are considered to form a futile cycle, which may consequently impair cells and cause APAP hepatotoxicity [150,151]. Consistent with this mechanism, the development of anion gap metabolic acidosis is likely associated with 5-oxoprolinuria, which is a symptom sometimes observed in patients taking APAP [152]. On the other hand, due to the broad substrate specificity in the γ-GCS reaction, instead of Cys, any amino acid could serve as the substrate for γ-glutamylation with varying efficiency. As a result, the auto-cyclization of γ-glutamyl phosphate is prevented by the presence of such an amino acid, which prevents progression of the futile cycle and consequential metabolic acidosis. This preventive mechanism is particularly effective when amino acids with high affinity to γ-GCS are present, and, hence, 2AB is considered to play such a role. OPT is detectable frequently in non-survivors of APAP-induced liver failure. Mean OPT levels are not associated with survival [153], however, which suggests that the production of OPT may not sufficiently suppress a futile cycle. To end this debate, careful consideration based on appropriate experiments performed on animal models is required.

A benefit has also been proposed in terms of the inhibition of ferroptosis by forming a γ-glutamyl peptide; in other words, the γ-glutamyl peptide-producing reaction is exploited to reduce cellular Glu content, which attenuates the production of ROS so as to suppress the lipid peroxidation that is responsible for cell death [154]. Cys starvation commonly causes ferroptosis in many types of cells through glutathione deprivation, whereas cells that abundantly produce γ-glutamyl peptides are resistant to nutritional deficiency. It is well established that a Cys deficiency decreases glutathione production, which precludes the GPX4-mediated reductive detoxification of lipid peroxides [30]. Also, the formation of γ-glutamyl peptides by γ-GCS consumes cellular Glu, which is converted to 2-ketoglutarate and becomes an intermediary compound to the tricarboxylic acid (TCA) cycle. As a result of the formation of γ-glutamyl peptides, the carbohydrate catabolism in the TCA cycle is attenuated. While the production of ATP is decreased by this, and the production of oxygen radicals that cause lipid peroxidation is also simultaneously decreased. Consequently, ferroptosis that is executed by lipid peroxidation is suppressed. This mechanism is supported by observations wherein inhibition of the electron transfer chain suppresses the ferroptosis caused by Cys deprivation [155,156]. A brief explanation is that a decline in the lipid peroxidation reaction due to attenuated Glu catabolism is a likely mechanism for the inhibition of ferroptosis in cells with a high capacity to produce γ-glutamyl peptides. Because each type of cell depends on a different type of metabolism, it is likely that the functions of γ-glutamyl peptides may also differ from cell to cell.

### 6.3. Extracellular Signaling Mediated by γ-Glutamyl Peptides Produced by GGT

Another question that also remains unanswered is that of why the γ-glutamyl group in glutathione is metabolized differently by extracellular GGT and intracellular CHAC. A variety of γ-glutamyl peptides have been identified in blood plasma and in some organs, including the liver and brain, and their numbers are elevated under pathological conditions or during malnutrition [140,142]. The stabilization of peptides by attaching the γ-glutamyl moiety is one of the purposes of the N-terminal modification, but the potential physiological action of regenerated γ-glutamyl peptides by GGT has long been debated.

Taurine (Tau) exerts pleiotropic activity in the central nervous system and in other organs [157]. Among the γ-glutamyl peptides, γ-glutamyl taurine (γ-Glu-Tau) has been studied to some extent compared to other γ-glutamyl peptides [158]. A recent study positively correlated accelerated aging with Tau deficiency in animals, including worms, mice, and monkeys [159], although the details of the mechanism remain unclear. Since Tau has no carboxyl group, most of it can exist in the free form, but exceptionally, it constitutes the second amino acid in dipeptides, as seen in γ-Glu-Tau, which is a representative γ-glutamyl peptide that is reportedly produced in the brain by GGT [160]. Our results, which are based on metabolite analyses of the tissues and plasma of mice as well as on cultivated cells, indicate that γ-GCS may also be responsible for the production of γ-Glu-Tau. This conclusion seems clear despite the high Km of Tau found in the γ-GCS reaction [140]. The proposed roles of γ-Glu-Tau include interactions with excitatory amino acidergic neurotransmission [161] and anti-epileptic activity [162]. How γ-Glu-Tau exerts such functions largely remains ambiguous, however, partly because the target receptor molecules remain unidentified [158].

It is conceivable that GGT hydrolytically produces Glu from glutathione when there is not enough amino acid present, and when sufficient amino acids are present, GGT transfers the γ-glutamyl moiety to the amino acid to generate γ-glutamyl peptides (Figure 8). In the brain, extracellular Glu content is maintained at extremely low levels because Glu may exert excitotoxicity in some neurons expressing NMDA-type glutamate receptors [163]. Therefore, the transfer of the γ-glutamyl moiety to other amino acids rather than the release of Glu could be advantageous as it can prevent the excitatory cytotoxicity of extracellular Glu—notably in injured brain tissue. Although this mechanism remains hypothetical, studies on xCT-knockout mice, which are unable to export cellular Glu in exchange for extracellular cystine, suggest a reduction in excitotoxicity [45,164] and indirectly support this hypothetical mechanism.

The discovery that glutathione binds to the G-protein-coupled calcium-sensing receptor (CaSR) [165] may provide a clue to understanding the extracellular production of γ-glutamyl peptides by GGT. Other γ-glutamyl peptides, which include GSSG and a mixed disulfide of Cys and GSH (CySSG), can also bind this unique receptor and modulate its function [166]. CaSR is systemically expressed in the brain and intestine, and appears to maintain extracellular calcium ion levels within a physiological range (1.1–1.3 mM) by regulating the secretion of parathyroid hormones [167]. CaSR possesses binding sites for multiple ligands, which include orthosteric agonist, extracellular Ca^2+^, and many allosteric compounds, such as amino acids and peptides that include γ-glutamyl peptides [10,166]. These properties of CaSR have particularly attracted research in the field of food chemistry because GSH and γ-Glu-Val-Gly act as “kokumi” taste substances that enhance sweetness, saltiness, and “umami” tastes without producing a taste of their own [168]. The gastro-intestinal tract expresses CaSR that mediates the actions of these γ-glutamyl peptides and leads to dietary hormone release in response to nutrients within the intestinal lumen [8,9].

Moreover, CaSR is reportedly involved in neuronal growth, migration, differentiation, and neurotransmission [167,169]. CaSR may also play a critical role in the central neuronal system under pathological conditions such as ischemia, Alzheimer’s disease, and in neuroblastoma [170]. Soluble Aβ reportedly binds CaSR and leads to cell death by aggravating neuronal inflammation [171,172]. Given these functions of CaSR, GSH and other γ-glutamyl peptides that are produced by means of GGT may modulate CaSR function, which helps protect against pathogenic stress.

Recent studies imply pharmacological benefits for γ-Glu-Cys in inflammation [173,174,175,176], stroke [177], ALS [178], and ischemia/reperfusion injury [179]. The increased production of GSH appears to rationalize the pharmacological action of γ-Glu-Cys [146,147,148]. In this scenario, GGT would remove the γ-glutamyl moiety and release Cys for de novo glutathione synthesis. However, since GGT expression has been observed only in a limited number of cells, there may be an alternative explanation for γ-Glu-Cys action. Upon the binding of γ-Glu-Cys to the CaSR pocket, allosteric activation could occur in intracellular signaling, because it leads to the suppression of inflammation in colitis mouse models [173]. Thus, CaSR could at least partially account for the beneficial action of γ-Glu-Cys, although this hypothetical mechanism must be verified in experiments that employ model animals such as mice with a genetic ablation of CaSR.

If CaSR binds extracellular γ-glutamyl peptides and modulates cellular activity in tissues, it would be interesting to determine the messages that are conveyed via γ-glutamyl peptides. In order to address this issue, research must consider the circumstances under which γ-glutamyl peptides are produced. γ-GCS produces γ-glutamyl peptides when the supply of cellular Cys is insufficient, such as in cases of excessive consumption of glutathione for conjugation or under the excessive production of peroxides due to oxidative stress [180]. Therefore, the cells in such a situation could produce γ-glutamyl peptides in order to transmit signals to surrounding cells to properly prepare for, or respond to, such an adverse situation. On the other hand, GGT tends to produce γ-glutamyl peptides when extracellular amino acids, as well as glutathione, are abundant. GGT1 on a brush-border membrane may not experience such a situation under healthy conditions because primary urine contains low levels of amino acids. GGT is known to be induced under pathological conditions, such as liver damage due to excessive alcohol intake or during tumor development. In cases of cell injury, damaged hepatocytes release damage-associated molecular patterns (DAMPs), which include amino acids and proteins [181]. Accordingly, GGT may be able to produce more γ-glutamyl peptides by utilizing the components of DAMPs in such situations rather than in a healthy state. As a result, CaSR-expressing cells could respond to pathological situations and prepare defense systems by inducing protective genes. This hypothetical mechanism also must be confirmed by experiments.

## 7. Perspectives

Glutathione is a pivotal molecule that protects cells from the cytotoxic effects of xenobiotic compounds via conjugation reactions and oxidative insults through the reductive detoxification of peroxides by donating electrons to the redox system. The degradation of glutathione by GGT recruits amino acids, notably Cys, to cells, but also produces γ-glutamyl peptides extracellularly under certain circumstances. γ-Glutamyl peptides are also produced intracellularly by γ-GCS-mediated reactions. These reactions appear to proceed under pathological conditions and act as a cellular defense. Although the significance of extracellularly produced γ-glutamyl peptides by GGT have long been obscured, the discovery that γ-glutamyl peptides modulate CaSR function has unveiled their potential roles in extracellular signaling. At present, the signaling role of γ-glutamyl peptides is known to be primarily manifested in the gastrointestinal tract and in the nervous system, but the systemic expression of CaSR may extend this phenomenon to other physiological functions, such as those in the cardiovascular and immune systems.

## Figures and Tables

**Figure 1 cells-12-02831-f001:**
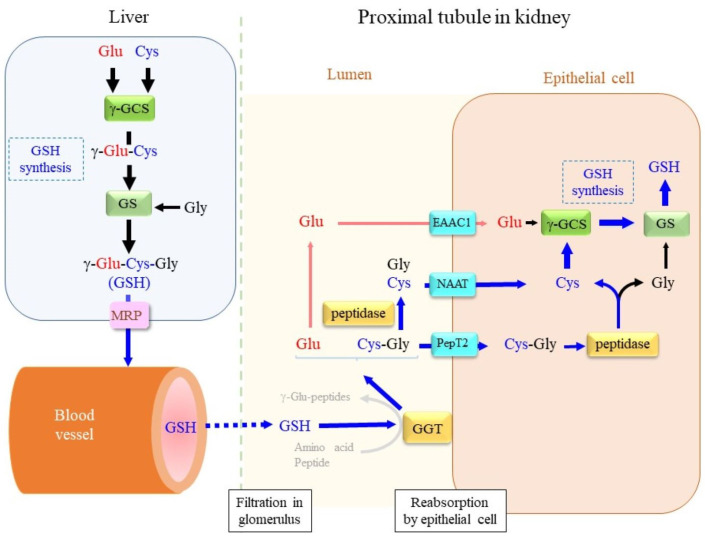
Glutathione dynamics in vivo. In the liver, glutathione is synthesized from the constituent amino acids Glu, Cys, and Gly by means of the sequential reactions of γ-GCS and GS under normal physiological conditions. The resultant glutathione is largely released into the bloodstream from cells via the multidrug resistance-associated protein (MRP) family. At the renal glomerulus, GSH as well as other nutrients are filtered mostly into the primary urine. The tubular epithelium is rich in GGT, which degrades glutathione. The γ-glutamyl moiety is removed by GGT, and the remaining Cys-Gly is either transported into cells directly via dipeptide transporter PepT2 or further hydrolyzed to Cys and Gly by dipeptidase, and then, transported via the neutral amino acid transporter NAAT. The γ-glutamyl moiety of glutathione is converted to free Glu, which is transported via EAAC1. Incorporated amino acids or dipeptides could contribute to glutathione synthesis. The production of new γ-glutamyl peptides by GGT is unlikely, however, because the extracellular contents of amino acids are limited to serving as substrates for the γ-glutamylation reaction.

**Figure 2 cells-12-02831-f002:**
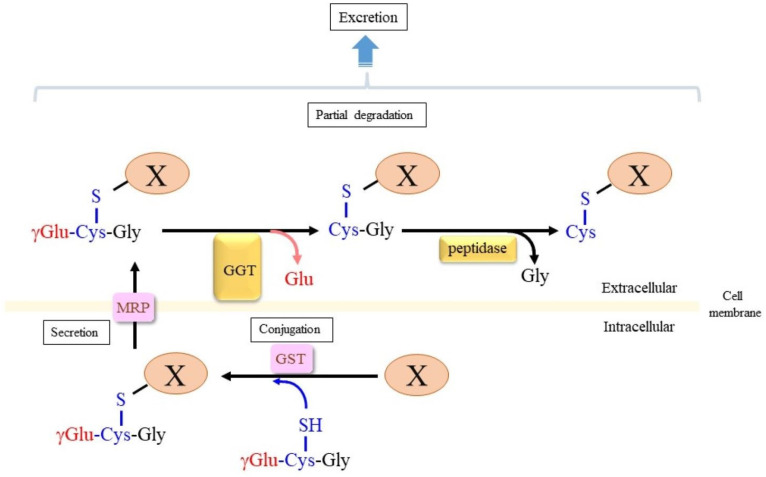
Glutathione conjugation of xenobiotic compounds and their metabolic conversion in the extracellular space. Xenobiotic compounds (X) experience metabolic detoxification mostly in the liver. Glutathione conjugation is one of three popular conjugation reactions that stimulate the secretion of conjugated compounds from the cell via MRP. GGT localized on the cell surface then removes the γ-glutamyl moiety, and then, the Gly unit can also be removed by extracellular dipeptidases. As a result, only the Cys portion of glutathione remains bound to a xenobiotic compound.

**Figure 3 cells-12-02831-f003:**
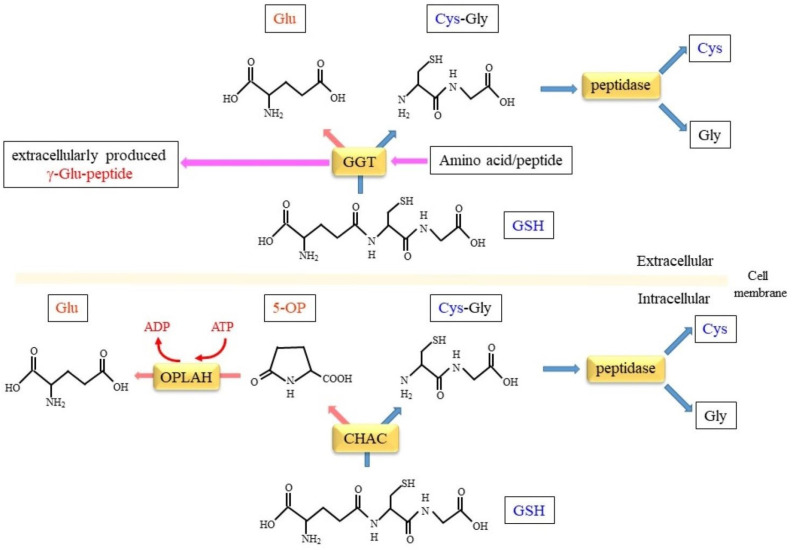
Comparing the catalytic reactions of GGT and CHAC. GGT in the extracellular space removes the γ-glutamyl moiety of glutathione, which results in either a Glu and Cys-Gly dipeptide by means of hydrolytic activity or a new γ-glutamyl peptide by means of transferase activity. The γ-glutamyl moiety of GSH is converted to L-5-oxo- proline (5-OP) by means of the aminoacyltransferase activity of CHAC. In an ATP-dependent manner, 5-OP is then hydrolyzed to Glu by 5-oxoprolinase (OPLAH).

**Figure 4 cells-12-02831-f004:**
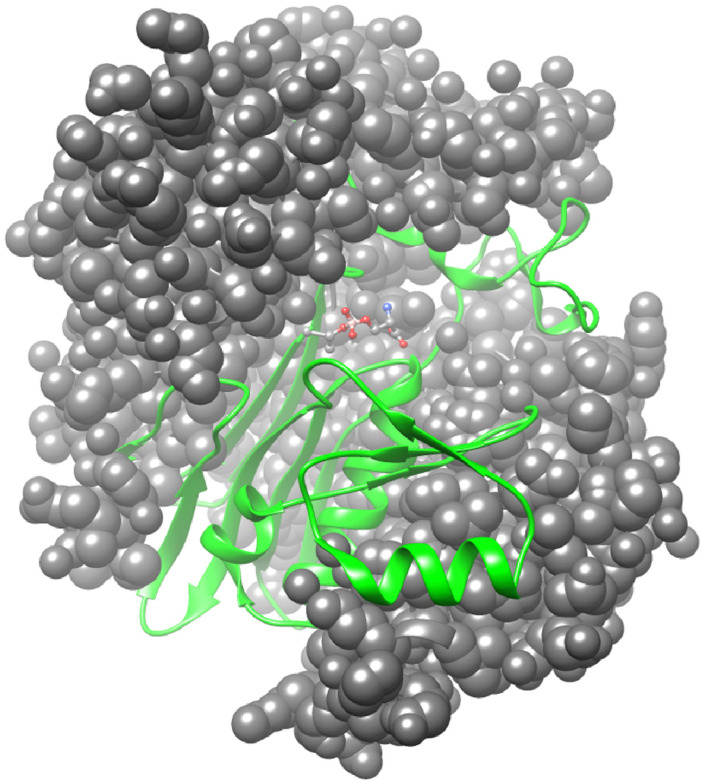
Molecular structure and dimeric association of the subunits of GGT. The large and small subunits are shown as a gray sphere model and a green ribbon model, respectively. The serine–borate complex as a transition state analog is indicated as a ball-and-stick model in the center of the molecule. The structure based on PDB ID: 4ZC6 was drawn using UCSF Chimera.

**Figure 5 cells-12-02831-f005:**
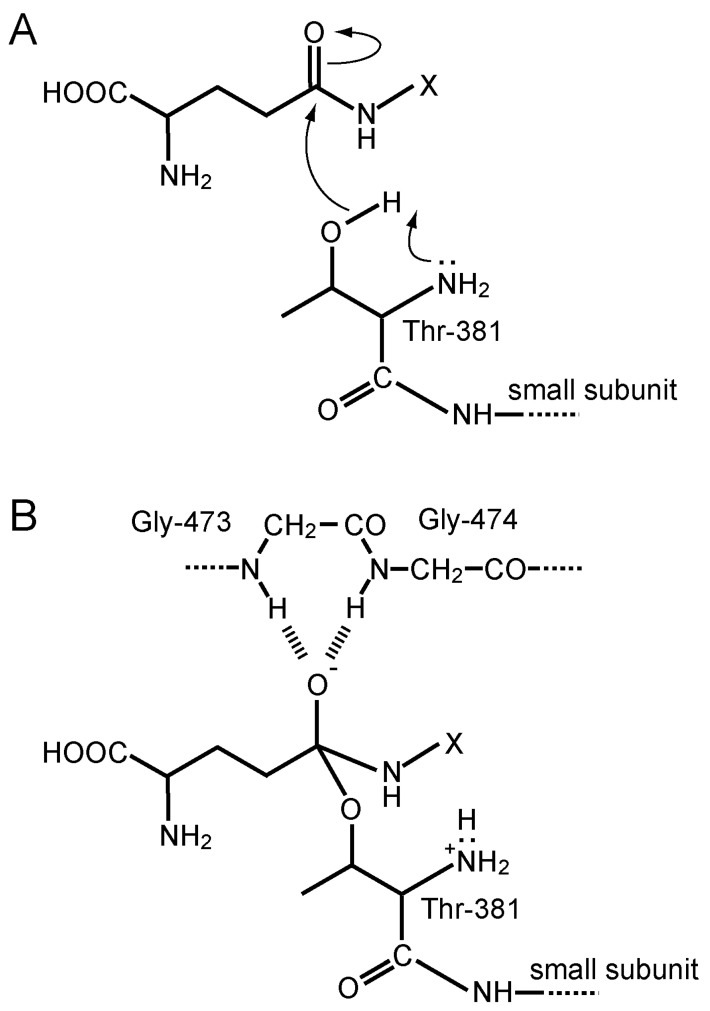
Catalytic mechanism of GGT. (**A**) Nucleophilic attack by the OH group of Thr-381 on the carbonyl group of the γ-glutamyl moiety of a substrate. The attack is assisted through the general base catalysis by the amino group of the same residue. (**B**) The transition state structure. The oxyanion of the tetrahedral intermediate may be stabilized by the hydrogen bonds with Gly-473 and Gly-474.

**Figure 6 cells-12-02831-f006:**
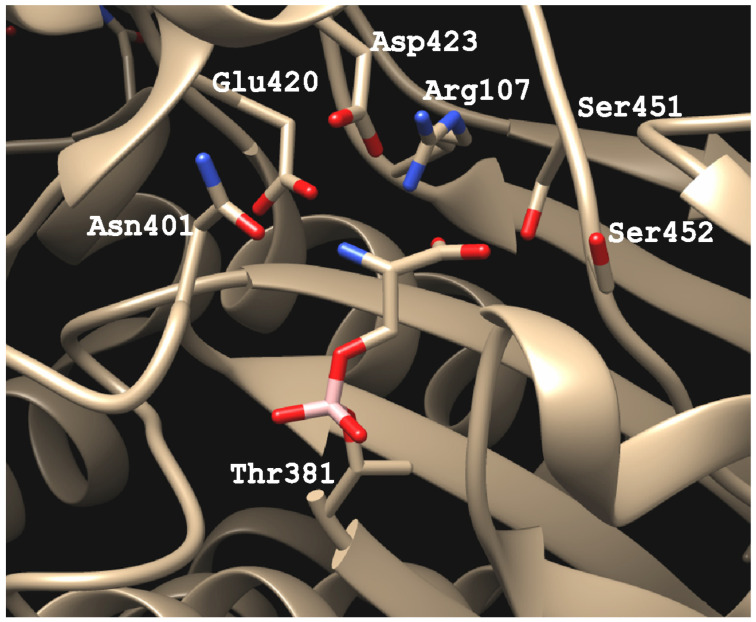
Active-site residues involved in substrate binding and catalysis. Thr-381 serves as a catalytic nucleophile. The other residues play roles in the binding of the γ-glutamyl group of a donor substrate, as described in the text. The serine–borate complex mimics the tetrahedral intermediate of the γ-glutamyl group. The OH group of the serine as a ligand and the catalytic OH group of Thr-381 are bridged by the borate anion. The structure based on PDB ID: 4ZC6 was drawn using UCSF Chimera.

**Figure 7 cells-12-02831-f007:**
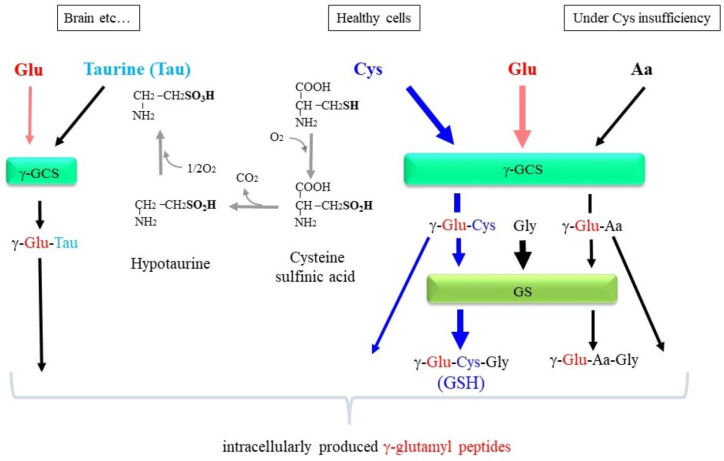
γ-Glutamyl peptides may be produced through the glutathione-synthesizing pathway. Cells that express both γ-GCS and GS synthesize glutathione in the presence of a sufficient amount of Cys. Under Cys deficiency, other amino acids (Aa) may be used instead of Cys by γ-GCS, which produces a variety of γ-glutamyl peptides. When GS activity is low, γ-glutamyl amino acids (γ-Glu-Cys and γ-Glu-Aa) are produced. Tau is produced from Cys through sequential oxidation reactions. Since Tau does not carry a carboxyl group, γ-Glu-Tau is the final product even in the presence of GS.

**Figure 8 cells-12-02831-f008:**
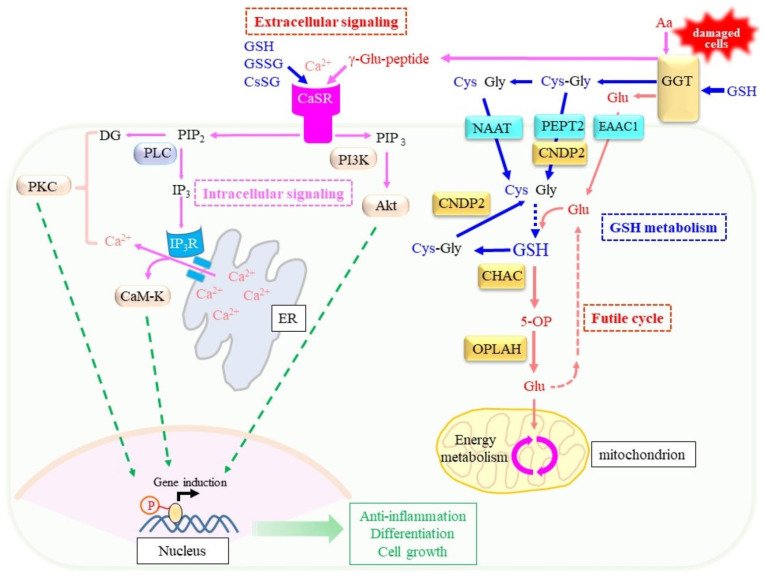
Extracellular γ-glutamyl peptides may exert extracellular signaling through allosteric modulation of CaSR. Two types of actions of GGT are shown: the hydrolysis of glutathione and production of γ-glutamyl peptides. CaSR is a G-protein-coupled receptor for extracellular calcium ions (Ca^2+^). A variety of ligands, including GSH, GSSG, Cys-linked glutathione (CsSG), and γ-glutamyl peptide (γ-Glu-peptide), allosterically bind CaSR. Activated CaSR may stimulate the conversion of phosphatidylinositol in the plasma membrane to inositol 1,4,5-trisphosphate (IP_3_) and diacylglycerol (DG). The receptor for IP_3_ (IP_3_R) is the Ca^2+^ channel located in the ER membrane. Released Ca^2+^ from the ER lumen together with DG activate protein kinase C (PKC). Ca^2+^ may bind calmodulin, and then, activate calmodulin-dependent protein kinase (CaM-K). On the other hand, the stimulation of phosphoinositide 3-kinase (PI3K) by CaSR results in phosphatidylinositol-3,4,5-trisphosphate (PIP_3_), which activates protein kinase B (Akt). These second messengers collectively stimulate the expressions of genes, which results in a variety of cellular responses, including cell growth, differentiation, and anti-inflammation.

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
