# Peer review of "The Emerging Roles of γ-Glutamyl Peptides Produced by γ-Glutamyltransferase and the Glutathione Synthesis System"

_cells, 2023, doi:10.3390/cells12242831_

Round 1
Reviewer 1 Report
Comments and Suggestions for Authors
In this article, Ikeda and Fujii review the current knowledge regarding the glutathione synthesis system in animals with a special emphasis on gamma-glutamyl peptides resulting from the action of gamma-glutamyl transferase. After an introduction on the synthesis of glutathione and its degradation, the authors describe its pleiotropic functions: xenobiotic metabolism, synthesis of bioactive compounds, redox homeostasis. How glutathione is maintained within the cells is then addressed. Next, the contribution of gamma-glutamyl transferase in the metabolism of glutathione is detailed by the authors. This enzyme is thus quite thoroughly detailed in terms of properties, structure, enzyme reactions, mechanism, gene. The authors then further describe the enzymatic reactions related to intracellular metabolism (gamma-glutamylcyclotransferase, 5-oxoprolinase, dipeptidases). Gamma-glutamyl peptides produced by gamma-glutamyl-cysteine synthetase and glutathione synthetase in normal and pathologic conditions are also discussed. Hypothesis regarding the benefits of the production of these peptides devoided of redox activities within the cells are presented in the article. The extracellular contribution of these peptides is also addressed, especially in relation to the binding of glutathione and other gamma-glutamyl peptides to the G-protein-coupled calcium-sensing receptor CaSR. Overall, this review article quoting 179 articles including very recent one is well written and useful in the field of glutathione research in general and gamma-glutamyl peptides in particular. I have only the following minor suggestions to consider:
1/ Line 228: you way quote the original reference for the inhibitor buthionine sulfoximine
2/ Lines 286-292: it may be relevant to further indicate that on the other end a mammalian mitochondrial glutathione transporter has been identified (Wang et al. (2021) Nature 599(7883):136-140. doi: 10.1038/s41586-021-04025-w).
3/ Line 310: correct “gamma-.Glutamyl tranferase” to “gamma-Glutamyl tranferase” (remove “.”)
4/ Lines 415-418 (Legend to Figure 5): you state that (B) is the transition state structure. Are you sure this is a transition state that is represented and not an enzyme intermediate?
5/ Line 611: correct “gamma-.GCS/GS” to “gamma-GCS/GS” (remove “.”)
Author Response
We have included our responses after the reviewer's comment.
1/ Line 228: you way quote the original reference for the inhibitor buthionine sulfoximine
Our responses: Thank you for kind advice. We have now quoted the original reference [new 54] for buthionine sulfoximine.
2/ Lines 286-292: it may be relevant to further indicate that on the other end a mammalian mitochondrial glutathione transporter has been identified (Wang et al. (2021) Nature 599(7883):136-140. doi: 10.1038/s41586-021-04025-w).
Our responses: As suggested by the reviewer, we have appropriately modified and added the descriptions on transport and mitochondrial import of GSH with the reference [new 69].
3/ Line 310: correct “gamma-.Glutamyl tranferase” to “gamma-Glutamyl tranferase” (remove “.”)
Our responses: We have corrected the enzyme name for GGT, as indicated by the reviewer.
4/ Lines 415-418 (Legend to Figure 5): you state that (B) is the transition state structure. Are you sure this is a transition state that is represented and not an enzyme intermediate?
Our responses: We are sure that (B) is the transition state structure. In the enzymes that involve covalent catalysis, as also seen in serine class hydrolases, a tetrahedral structure is reasonably assumed between the enzyme/substrate complex and the acyl enzyme intermediate in the course of reaction, and it is generally regarded and accepted as a transition state.
5/ Line 611: correct “gamma-.GCS/GS” to “gamma-GCS/GS” (remove “.”)
Our responses: Thank you very much for pointing out. We have removed “.” after “γ-“.
Reviewer 2 Report
Comments and Suggestions for Authors
The work concerns glutathione - its function and its metabolism, with a particular focus on GGT, and the potential roles of the γ-glutamyl peptides that are extracellularly produced by GGT.
In the extensive section on glutathione (chapters 2, 3 and 4) there is a large amount of very basic information regarding the commonly known functions and metabolism of glutathione - proportionally too much in relation to the volume of chapter 6, which contains the information signaled in the title.
If we take into account the part of the work that presents the roles of γ-glutamyl peptides produced by γ-glutamyl transferase, this work would not meet the requirements for review work in a journal such as Cells.
Author Response
Our responses: Thank you for your critical comments regarding description on mechanistic aspects of γ-glutamyl transferase. As the reviewer accurately points out, this review focuses on the novel features of γ-glutamyl peptides. γ-Glutamyl peptides are synthesized both by γ-glutamyl cysteine synthetase (γ-GCS) inside cells and by γ-glutamyl transferase (GGT) outside cells. As mentioned in the text, excellent review articles that have overviewed the molecular mechanism of γ-glutamyl peptides synthesis by γ-GCS are available. However, despite popularity of GGT as a marker for diseases and as a unique producer of the extracellular γ-glutamyl peptides, there is no suitable literatures that overview recent advance concerning molecular mechanism of their production by GGT. Moreover, from an enzymatic perspective, GGT has the unique property; i.e. it either hydrolyzes γ-glutamyl moiety or transfers it to peptides. Since this enzymatic property is deeply associated to the synthesis of γ-glutamyl peptides as extracellular signaling molecules, it is essential to know its catalytic mechanism for the purpose of understanding physiological function of γ-glutamyl peptides. We believe it is also a great advantage that readers can learn about both the producing mechanism and physiological functions of γ-glutamyl peptides, just by reading this article containing corresponding chapters.
Reviewer 3 Report
Comments and Suggestions for Authors
The manuscript is clear, relevant for the field and presented in a well-structured manner. The comprehensiveness is visible, so every reader will have the opportunity to repeat the knowledge about glutathione as a very important endogenous antioxidant, but also an insight into the latest knowledge about the maintenance of redox homeostasis and the application of intercellular messaging.
The authors presented comprehensive current knowledge on functions and metabolism of glutathione as very important molecule in organism. Although there are very good review papers on GSH, the approach of these authors is focused on roles of γ-glutamyl peptides that captures the current hot topics of scientific research like ferroptosis which can be seen from the most recent cited references (reference number 33-38). In general, the paper is easy to read, although it offers a lot of information and details about each segment: synthesis, degradation, transport of GSH, etc. It is commendable that the authors provide detailed instructions to references for everything that is already known, from which more information can be found. Furthermore, the authors give new perspectives by trying to provide answers to various intriguing questions i.e. whether gamma glutamyl peptides produced by different enzymes have different roles regarding their signaling role. This review is written in an appropriate way according to Instructions for Authors, therefore I would suggest that the paper can be accepted without any further changes.
Author Response
Our responses: We would like to thank the reviewer for understanding our intentions on writing this article and providing favorable comments.
Round 2
Reviewer 2 Report
Comments and Suggestions for Authors
The work has not been changed, even in the first chapters, so I stand by my decision - I am not convinced by the authors' explanations.